# Cell Therapy for Neurological Disorders: The Perspective of Promising Cells

**DOI:** 10.3390/biology10111142

**Published:** 2021-11-06

**Authors:** Donghui Liu, Larisa Bobrovskaya, Xin-Fu Zhou

**Affiliations:** School of Pharmacy and Medical Sciences, University of South Australia, Adelaide, SA 5000, Australia; liudy016@mymail.unisa.edu.au (D.L.); Larisa.Bobrovskaya@unisa.edu.au (L.B.)

**Keywords:** cell therapy, cell transplantation, neurological disorders, stem cells, clinical trials

## Abstract

**Simple Summary:**

Cell therapy has become a powerful method for regenerative medicine. However, there has not been an ideal cell type and cell source for the treatment of neurological diseases such as Parkinson’s disease and Alzheimer’s disease. This review aims to introduce the potentials of different cells for treating neurological disorders by collecting the results from related clinical trials and recent animal studies. It is an overview of some promising cells that may be clinically used for neurological disorders. The characteristics of each cell type and the main mechanism of function are also described.

**Abstract:**

Neurological disorders are big public health challenges that are afflicting hundreds of millions of people around the world. Although many conventional pharmacological therapies have been tested in patients, their therapeutic efficacies to alleviate their symptoms and slow down the course of the diseases are usually limited. Cell therapy has attracted the interest of many researchers in the last several decades and has brought new hope for treating neurological disorders. Moreover, numerous studies have shown promising results. However, none of the studies has led to a promising therapy for patients with neurological disorders, despite the ongoing and completed clinical trials. There are many factors that may affect the outcome of cell therapy for neurological disorders due to the complexity of the nervous system, especially cell types for transplantation and the specific disease for treatment. This paper provides a review of the various cell types from humans that may be clinically used for neurological disorders, based on their characteristics and current progress in related studies.

## 1. Introduction

Neurological disorders are defined as diseases of the nervous system. There are more than 600 neurological disorders that can be divided into four categories: (1) sudden onset conditions, such as stroke; (2) intermittent conditions, such as epilepsy; (3) progressive conditions, such as Parkinson’s disease (PD); (4) stable neurological conditions, such as cerebral palsy. These diseases have become a big public health challenge and present a tremendous burden to individuals, their families, and society. Some of the disorders can be common and severe, including stroke, Huntington’s disease (HD), traumatic brain injury (TBI), spinal cord injury (SCI), epilepsy, PD, Alzheimer’s disease (AD), Lyme disease, cerebral palsy, ataxia, amyotrophic lateral sclerosis (ALS), hypoxic-ischemic encephalopathy (HIE) and multiple sclerosis (MS). Currently, although there are many conventional therapies that can alleviate their symptoms, such as levodopa or deep brain stimulation for PD [1], cholinesterase inhibitors, or memantine for AD [2], these treatments have failed to slow or reverse the progression of the diseases, in other words, these diseases are not curable by the conventional therapies.

A common feature of many severe neurological disorders is the loss and/or dysfunction of massive neural cells, especially neurons. Therefore, regenerative medicine, especially cell therapy, has become an intriguing field for researchers in the last several decades. Cell therapy is thought to be an excellent therapy for many neurological disorders acting by replacing dead cells or/and releasing protective factors to the damaged cells or/and modulating the lesion’s microenvironment in the nervous system [3,4,5]. With the rapidly expanding studies in this research area, cell therapy has shown its unique potentials in treating neurological disorders. Currently, various types of cells have been used in cell therapy studies, including embryonic stem cells (ESCs), induced pluripotent stem cells (iPSCs), neural stem/progenitor cells (NSPCs), mesenchymal stromal/stem cells (MSCs), and olfactory ensheathing cells (OECs), etc. However, despite the promising results from numerous animal-based studies, the outcomes from clinical trials are normally not as significant as from animal studies. This may be due to the difference between animal models and the real disease of patients and the differences in nervous systems between animals and humans.

The outcome of cell therapy can be affected by many factors, including cell source, cell type, route of administration, and the target disease. The aim of this review is to summarize the cells that may be clinically used for neurological disorders, especially for the diseases mentioned above, and their characteristics and current progresses in related studies.

## 2. Cell Source: Autologous or Allogeneic?

The transplantable cells for patients can be divided into two sources: autologous and allogeneic. Autologous cells are usually more favorable than allogeneic cells in regenerative medicine as they avoid several issues that allogeneic cells may have, such as immune rejection or the use of immunosuppressants, ethical issues, and finding suitable donors [6]. However, allogeneic transplants are necessary for the cells that are therapeutic but hard to obtain from the patients themselves, such as ESCs [7], NSPCs [8], and fetal stem cells (FSCs) that contain various stem cells from fetal blood and tissues (e.g., MSCs, NSPCs, and hematopoietic stem cells) (see Table 1). Nowadays, cell reprogramming provides a new opportunity to generate any type of autologous cells from another type of easily obtainable somatic cells, such as iPSCs from urine [9] and induced neural stem cells from fibroblasts [10]. However, the safety of these cells is highly related to the methods and vectors used in induction protocols [11,12,13], and the safety and therapeutic effects need to be well studied before taking into clinical application.

Interestingly, some cells, such as umbilical cord blood mononuclear cells and adipose-derived mesenchymal stem cells (hADSCs), have been reported as immature immune cells; namely, they will not cause immune rejection, even in the absence of immune suppression [14,15], which may make them suitable candidates as allogeneic transplants.

## 3. Different Cell Types and the Current Progress

### 3.1. Embryonic Stem Cells (hESCs)

Human ESCs (hESCs) are pluripotent cells that are derived from human embryos and are capable of self-renewing and differentiating into all types of human cells. The first cell line of hESCs was isolated by James Thomson in 1998 [16], which then brought a new hope of developing new hESC lines for cell therapy. The therapeutic potential of ESCs for neurological disorders was proved a long time ago [17,18]. However, several critical issues of hESCs have limited the related studies and clinical applications: (1) ethical issues, as the establishment of hESCs lines involves the exploitation and destruction of human embryos; (2) immune rejection. As hESCs are allogeneic cells, they may cause immune rejection after transplantation [19], or the patient may need a life-long administration of immunosuppressants; (3) tumor formation. Due to the pluripotency of hESCs, they have the potential to generate teratoma after transplantation [20,21]. Several methods have been reported to reduce or eliminate the tumorigenicity of ESCs, such as pre-treating ESCs with mitomycin [22], co-transplantation ESCs with MSCs [23]. Among these methods, pre-differentiating ESCs into target cells prior to transplantation is a favorable way. hESCs are normally taken as a source of different cells in vitro. hESCs derivates, such as hESCs-derived oligodendrocytes, hESCs-derived dopaminergic neurons, have shown their therapeutic effects in animal studies for treating neurological disorders [24,25]. Although several animal studies transplanting hESCs for neurological disorders (such as SCI [26]) have shown some therapeutic effects, hESCs are normally considered unlikely to be directly grafted for cell therapy. Remarkably, however, Geeta Shroff has transplanted hESCs via a similar cell delivery method into patients with Lyme disease, multiple sclerosis, spinal cord injury, stroke, or cerebral palsy, and has proved the effectiveness and safety of the cells (see Table 2), with some ethical concerns [27,28]. Moreover, it has been claimed that these hESCs prepared at their institute are unlikely to cause immune rejection even without immunosuppressants as they were harvested at the very initial stage of blastocyst when the genesis was not activated [7]. To summarize, hESCs are more used as a source of other transplantable cells rather than a direct transplantation candidate in cell therapy studies. However, a certain line of hESCs may become a suitable candidate for cell therapy for neurological disorders if the critical issues can be overcome in the future.

### 3.2. Induced Pluripotent Stem Cells (iPSCs)

The first human iPSCs (hiPSCs) line was generated by Takahashi et al. [35] after they introduced four factors (Oct3/4, Sox2, Klf4, and c-Myc) into human fibroblasts. hiPSCs share similarities with hESCs in many aspects such as proliferation, morphology, gene expression, and differentiation. This iPSCs technique (also called cell reprogramming) provides an opportunity to generate any type of patient-specific cells from other obtainable cells, such as fibroblasts or urine cells. Therefore, there are no ethical issues from hiPSCs. Moreover, autologous cells transplantation from patients themselves has no risk of immune rejection and no need to use immunosuppressants. However, there is still the risk of tumor formation [21]. iPSCs are classically generated by the integration of transcription factors with a viral vector, which may cause tumorigenesis and/or unpredictable mutagenesis in the genome. hiPSCs generated by non-integration methods, such as mRNA [36], plasmid [37], small molecules [38], are safer candidates for transplantation, although they are normally with comparatively low reprogramming efficiency [39].

To date, although several animal studies grafting hiPSCs have shown their therapeutic effects for neurological disorders, such as stroke and SCI [5,40] (see Appendix A), similar to hESCs, hiPSCs are also usually taken as a source of other transplantable cells in vitro. Pre-differentiated hiPSCs, including hiPSCs-derived NSPCs, hiPSCs-derived neurons and hiPSCs-derived MSCs have been shown effective for neurological disorders [41,42,43]. Especially, several clinical trials based on hiPSCs derivates have been launched for the treatment of PD and SCI [44,45], and thus have deeply encouraged the use of hiPSCs in treating neurological disorders. However, it is noteworthy that hiPSCs-derived NSPCs may still have the potential to generate tumors [46]. The safety issues, including tumorigenicity and aberrant reprogramming of the cells, need to be clearly addressed before their clinical application.

### 3.3. Neural Stem/Progenitor Cells (NSPCs)

NSPCs are multipotent cells that can self-renew and generate the main types of cells making up the central nervous system (CNS), including neurons, oligodendrocytes, and astrocytes. NSPCs have a lower risk of tumor formation compared to ESCs and iPSCs as they are more specialized cells and have less self-renewing ability. Therefore, human NSPCs (hNSPCs) are considered a favorable candidate for treating neurological disorders. Although it has been shown that a subset of NSPCs is present in highly restricted regions during adult life, their proliferation declines with aging [47,48], and it is nearly impossible to isolate autologous hNSPCs for cell therapy. The common sources of hNSPCs for research are human fetuses, hESCs, hiPSCs, or hMSCs. Like hESCs, hNSPCs directly isolated from embryos or differentiated from hESCs also have ethical concerns and may cause immune rejection, and hNSPCs from hiPSCs may have the same issues as hiPSCs as mentioned above. hNSPCs can also be obtained by trans-differentiation from hMSCs under specific experimental conditions [49]. In addition, hMSCs-derived hNSPCs are normally preferred in clinical studies as they normally have fewer issues than those from hESCs or hiPSCs. However, it is still important to generate a stable, efficient, and standardized protocol to convert hMSCs to hNSPCs and study the similarity between hMSCs-derived hNSPCs and bona fide hNSPCs.

Currently, hundreds of animal studies have used hNSPCs for treating various neurological disorders, including stroke, SCI, TBI, PD, HD (see Appendix A), which are far more than hESCs or hiPSCs transplantation studies. Moreover, their effectiveness has been proved and reported better than human MSCs in some conditions [50,51]. The grafted hNSPCs can not only differentiate into neurons and glia cells and establish a graft-host connection [52,53] but also produce trophic factors and modulate lesion microenvironment to improve behavior recovery [54,55]. Intriguingly, even the injection of hNSPCs secretome alone has been reported to support the functional recovery of 6-hydroxydopamine (6-OHDA) PD rats [56]. Many methods have been applied to enhance the therapeutic effects of hNSPCs, such as genetically modified hNSPCs by overexpressing a selected trophic factor. Brain-derived neurotrophic factor (BDNF)-overexpressing hNSPCs, glial cell line-derived neurotrophic factor (GDNF)-overexpressing hNSPCs, and insulin-like growth factor 1 (IGF-1)-overexpressing hNSPCs have all been shown significant therapeutic effects on neurological disorders [57,58,59]. In addition, pre-treating hNSPCs with a gamma-secretase inhibitor [60], metformin [61] or tumor necrosis factor α (TNFα) [62], co-transplantation of hNSPCs with MSCs [63] or using biomaterial scaffolds as a carrier for hNSPCs [64,65] have also been reported to improve their therapeutic potential for neurological diseases. However, it is worth noting that the subtype of hNSPCs may also influence their therapeutic effects for a specific disease. For example, the human fetal spinal cord-derived NSPCs or spinal cord-type NSPCs from hiPSCs have been shown to improve motor functions after SCI, but not human fetal brain-derived NSPCs or forebrain-type NSPCs from hiPSCs [64,66].

To date, some clinical studies have been accomplished, and a few have shown beneficial outcomes after hNSPCs transplantation for neurological disorders (see Table 3). For allogeneic transplantation, focusing on specific hNSPC lines may help to keep the consistency of the outcomes from bench to beside. Several clinical-grade hNSPC lines have been applied in clinical trials, including NSI-566. NSI-566 cell line has been shown to be safe and potential effective in ALS patients [67,68] and now is in a phase 3 trial for treating ALS. Moreover, the NSI-566 cell line has also been reported to significantly improve the behavioral and histological recovery of ischemic stroke patients [69]. On the other hand, hNSPCs from autologous MSCs are a preferred candidate for clinical therapies of neurological disorders and have been applied for treating cerebral palsy, MS and TBI, in clinical trials [70,71,72,73]. However, most of the trials are in phase 1 or 2 stages, which have proved the safety but not the statistical therapeutic effects of the cells. Further studies need to be performed to better understand the therapeutic effects of the cells for neurological disorders.

Induced neural stem/progenitor cells (iNSPCs), such as iPSCs, are generated from other types of somatic cells (such as fibroblasts or urine cells) but bypass the pluripotency stage. Generation of hiNSPCs is an attractive field and is more favorable than hiPSCs in treating neural diseases, as they are easier to differentiate into terminal neural cells and less tumorigenic. Currently, many hiNSPC lines have been generated through different methods [74,75], and several lines have been tested in animal disease models, such as SCI, glioblastoma, and stoke and have shown their significant therapeutic effects [76,77,78,79]. However, the safety and therapeutic potential of hiNSPC are highly related to the induction protocol and need to be further studied. Therefore, it is essential to establish a safe, stable, and efficient protocol to generate hiNSPC for cell therapy studies.

### 3.4. Neurons, Oligodendrocytes, and Astrocytes

Neurons, oligodendrocytes, and astrocytes are three main cell types that form the CNS and originate from a common lineage of hNSPCs during development. These cells are terminal cells from NSPCs differentiation and can also be obtained from human embryos, hESCs, hiPSCs, or hMSCs in cell therapy studies. Transplantation of these pre-differentiated cells can avoid unexpected or unwanted differentiation or tumor formation of stem cells in vivo, especially in neurological disorders that are associated with the loss or dysfunction of specific types of neural cells, such as transplantation of dopaminergic neurons or dopaminergic progenitors for PD. It has been reported that pre-differentiated GABAergic neurons from hNSPCs have shown greater repopulation of the damaged brain and better neurogenic activity and functional recovery than hNSPCs in a stroke model after transplantation, while hNSPCs have predominantly differentiated into astrocytes [86].

Because neurons are the main functional cells in the nervous system, they are usually more attractive than glia cells for regenerative medicine in neuroscience research. Studies have transplanted neurons or neuron progenitors into different disease models and have shown their therapeutic outcomes (see Appendix A). However, it is believed that more mature/differentiated donor cells have less survival capacity after transplantation [87,88,89]. Therefore, for a specific neuron type transplantation (for example, dopaminergic neurons), an immature stage of the cells may lead to better outcomes than mature cells or progenitors [89,90,91]. In addition, as the pre-differentiated cells have less proliferation capacity, they may need more cells for transplantation to reach a therapeutic level. Remarkably, several clinical trials transplanting neurons or neuron-contained tissue have proved the survival and potential therapeutic effects in patients (see Table 4). Moreover, direct reprogramming of human neurons from other somatic cells may also provide a suitable source of therapeutic neurons for cell therapies [92,93].

Oligodendrocytes and astrocytes are glia cells that are thought of as supportive and protective cells in the nervous system. A recent study has shown that the conditioned medium from hiPSCs-derived glial progenitor cells could result in better therapeutic effects than that from hiPSCs-derived neuronal progenitor cells and show a higher content of neurotrophins, indicating their potentials for cell therapy [94]. The main function of oligodendrocytes is to form myelin sheaths to support and insulate axons. Therefore, oligodendrocytes are highly related to diseases involving demyelination, such as MS and ALS. It has been suggested that transplantation of human oligodendrocyte precursor cells (hOPCs) could result in notable therapeutic outcomes in animals with MS or SCI [95,96,97], whereas transplantation of mature oligodendrocytes has failed to remyelinate naked axons in SCI [98,99]. Therefore, transplantation of hOPCs may be a potential therapy for demyelinating diseases. On the other hand, astrocytes are the most numerous cell type in the brain, playing various functions, including maintaining homeostasis, providing neurotrophic support, and connecting neurons with the bloodstream [100]. The loss or dysfunction of astrocytes is related to many neurological disorders, such as stroke, epilepsy, and MS [101]. Moreover, the astrocyte-formed glial scars are a reason that prevents neuroregeneration after CNS injury [102]. However, a study has also reported functional recovery after grafting hiPSCs-derived astrocytes with overexpressing the major glutamate transporter, GLT1, in an SCI model, suggesting their pro-regenerative function [103]. Notably, it is reported that the therapeutic effects of astrocytes are highly related to the subtypes of the cells. Stephen et al. found that astrocytes generated from human glial precursor cells by exposure to bone morphogenetic protein could promote significant functional recovery after SCI, whereas astrocytes generated by exposing the cells to ciliary neurotrophic factor failed to generate similar results [104]. Overall, the number of transplantation studies targeting oligodendrocytes or astrocytes is much lower than neuron-based studies. However, with better understanding of their functions in CNS, and more accessible sources of human oligodendrocytes or astrocytes (e.g., from hiPSCs), transplantation of oligodendrocytes or astrocytes may be a promising approach for treating specific neurological disorders.

### 3.5. Mesenchymal Stromal/Stem Cells (MSCs)

MSCs are multipotent cells that can differentiate into various cell types, including bone cells, muscle cells. In addition, it has been reported that MSCs can be transdifferentiated into neural cells under specific conditions, indicating the potential of MSCs for treating neurological diseases [49,108]. Human MSCs (hMSCs) can be easily obtained from many sources, among which the most studied hMSCs are those from bone marrow, umbilical cord, and adipose tissue. Although the detailed differences in their biological characteristics from different sources, the easy acquisition and the common properties, such as the capacity of self-renewing, multi-lineage differentiation, low tumorigenicity, low immunogenicity, and immunoregulatory function, have made hMSCs a promising candidate for cell therapy. Moreover, as mentioned above, hMSCs can also be a source of transplantable neural cells in vitro in models of neurological disorders.

Due to these advantages, hMSCs are currently one of the favorite cell types in cell therapy studies involving treating neurological diseases. Hundreds of animal studies have proved their safety and therapeutic effects for various neurological disorders (see Appendix A). Moreover, even hMSC-conditioned medium or hMSCs-derived exosomes have been shown to alleviate the symptoms of experimental stroke or SCI [109,110]. Notably, although it has been reported that hMSCs can differentiate into neural cells in the injured area after transplantation [111,112], other researchers have also found that hMSCs could result in significant functional recovery without neural differentiation or even when hMSC is cleared away at the end of the experiment, suggesting that the main mechanisms of their beneficial functions are neurogenesis and angiogenesis promotion, anti-apoptosis, anti-inflammation, and immunomodulation rather than cell replacement [113,114,115,116,117]. Furthermore, studies have demonstrated that hMSCs can cross the blood-brain barrier (BBB) and home to the injured site through intravenous administration, which is a more suitable route for clinical use compared to direct local delivery to the affected tissue [118,119]. However, it is noteworthy that negative outcomes from hMSCs transplantation have also been shown in several studies, suggesting that administration of hMSCs alone may not be enough to generate significant therapeutic outcomes in some cases [120,121]. Therefore, several methods have been used to improve their therapeutic functions, such as pre-treatment [122], genetic modification [123], co-transplantation with hNSPCs [63], combined with other treatments [124], and using biomaterial scaffolds as a carrier [125]. Interestingly, the therapeutic effects of hMSCs may also be related to the source of the cells or the age of the donor. Jumpei et al. have reported that human cranial bone-derived MSCs could result in significant functional recovery in a rat model of stroke, but not human iliac bone-derived MSCs [126]. Susumu et al. have reported that transplantation of hMSCs from young donors could provide better functional recovery through multiple mechanisms than old hMSCs [127].

To date, many clinical trials using umbilical cord-derived MSCs (UC-MSCs) or autologous bone marrow (BM-MSCs) or adipose-derived MSCs (hADSCs) for treating various neurological disorders have been conducted and published (see Table 5). However, most of the studies have only demonstrated the safety hMSCs administration. Although the therapeutic effects have been seen in many clinical trials, the number of patients in each trial is usually not sufficient for analyzing the efficacy. Therefore, it is necessary to take hMSCs to the next step using more patients and proper controls to study their therapeutic effects and the mode of action in specific neurological disorders.

### 3.6. Dental Pulp Stem Cells (DPSCs) and Stem Cells from Human Exfoliated Deciduous Teeth (SHED)

DPSCs and SHED are derived from the dental pulp of adult permanent teeth and baby deciduous teeth, respectively. They are ectoderm-derived stem cells originating from neural crest cells and possess similar characteristics as MSCs, including the capacity of self-renew and multi-lineage differentiation and expression of MSC-related markers. However, it is still controversial to define DPSCs and SHED as MSCs, mainly due to their different potency of differentiating into specific lineages [148]. Because of their easy accessibility by routine dental procedures and their MSC-like properties, DPSCs and SHED have gained more attention in the last decade in the field of regenerative medicine, including treating neurological diseases. Moreover, they can maintain their stemness and multipotency for many years by cryopreservation, therefore, providing an opportunity for cell banking [149,150].

Remarkably, it has been reported that, compared to BM-MSCs or hADSCs, DPSCs have a higher growth rate, stronger neurogenesis, and better neuro-supportive and neuro-protective properties in neurological injuries and pathologies [148], indicating they may have better therapeutic effects for neurological disorders. DPSCs have been used to treat many neurological disorders in animal studies and have led to significant beneficial outcomes (see Appendix A), mainly through the secretion of neurotrophic factors and anti-inflammatory functions [151]. It has been reported that the expression of neurotrophic factors in DPSCs is higher than that of hADSCs and BM-MSCs [152]. The conditioned medium from DPSCs can also improve several neuropathological conditions, including ALS [151,153]. On the other hand, the therapeutic effects of SHED for neurological disorders have also been studied in some animal studies, which have shown improved results (see Appendix A). It is suggested that SHED are in a more immature state than DPSCs with higher expression of pluripotent markers and a higher proliferation rate, while DPSCs show higher expression of neuroectodermal markers [154,155]. Similarly, the conditioned medium of SHED has also been proved to be therapeutic for animal model of stroke, PD, and TBI etc [156,157,158].

Overall, DPSCs and SHED are also promising candidates for treating neurological disorders and may be beyond hADSCs and BM-MSCs. However, due to the limited volume of the pulp tissue, it usually takes months to obtain enough cells for therapy from the primary isolation, although they have a high proliferation rate [148]. Moreover, the heterogeneity of DPSCs and SHED may affect their therapeutic potentials [159,160]. Moreover, most of the available evidence of their therapeutic function for neurological disorders was acquired using nonhuman xenotransplants. It is still a long way to study their therapeutic and side effects in humans.

### 3.7. Muse Cells

Muse cells (multilineage-differentiating stress enduring cells) are non-cancerous pluripotent stem cells that were first discovered by Yasumasa et al. in 2010 [161]. Muse cells are sporadically present in the connective tissue of nearly all organs, such as bone marrow and umbilical cord, and can even be collected from commercial cell lines, including human fibroblasts and bone marrow MSCs [161,162,163]. Interestingly, it has been reported that Muse cells are the primary source of hiPSCs in human fibroblasts, but not the non-Muse cells [164,165]. Moreover, as Muse cells are present in cultured MSCs, it is thought that Muse cells are more therapeutic in clinical MSCs therapies regarding tissue regeneration [166]. The characteristics of Muse cells, including non-tumorigenicity, pluripotency, and easy collection, have made Muse cells a promising candidate for cell therapy.

Several animal studies have shown the safety and effectiveness of human Muse cells for treating neurological disorders, such as stroke, intracerebral hemorrhage, encephalopathy, ALS, and SCI (see Appendix A). Moreover, Muse cells can engraft and integrate into the damaged regions and differentiate into neuronal cells, and finally, lead to functional and morphological recovery after intravenous administration [167,168,169,170]. However, Muse cells are still a novel type of stem cells that have not been well studied as other stem cells. With more related studies, we will obtain a better understanding of Muse cells and their potentials for treating neurological disorders.

### 3.8. Olfactory Ensheathing Cells (OECs)

OECs, also known as olfactory ensheathing glia, are terminally differentiated and self-renewable cells that are found in both the peripheral nervous system (PNS) and CNS, supporting the regeneration of the olfactory system throughout life. OECs can be isolated from the olfactory bulb (OB-OECs) through intracranial surgery or from olfactory mucosa (OM-OECs) through a simple, non-invasive nasal biopsy, which is preferred for autologous transplantation. Based on these properties, human OECs (hOECs) are considered a suitable candidate for CNS transplantation, particularly for SCI treatment. Numerous animal studies have shown that hOECs could promote the recovery from SCI through various mechanisms, including neuroprotection, promoting axonal growth/sprouting, improving angiogenesis, and restriction of glial scar [171,172]. Notably, OECs from different sites may act through different mechanisms. It has been reported that OM-OECs could regulate extracellular matrix and improve angiogenesis, while OB-OECs intend to improve axonal regeneration [172]. In addition to the animal studies, hOECs have also been used in clinical trials for SCI since the early 2000s (see Table 6). Although with a relatively small size of patients and variable outcomes in different patients, hOECs transplantation has led to different levels of functional recovery in most cases, indicating their therapeutic roles for SCI. Moreover, hOECs have also been used for treating ALS and cerebral palsy patients and have resulted in functional improvements, suggesting their therapeutic potentials are not only for SCI but also for other neurological disorders [173,174].

However, several bottlenecks are impeding the therapeutic effects of hOECs. hOECs isolated from the olfactory bulb or olfactory mucosa usually contain contamination cells such as fibroblasts that may affect their effects and need to be purified [175,176,177]. Moreover, the difficulty to quickly expand, poor survival after suspension injection, limited migration, and phagocytosis are also considered the major hampers of hOECs treatments [175]. Although various methods have been used to optimize OECs transplantation, it is still far from being a mature treatment for neurological disorders, including SCI [175].

### 3.9. Hematopoietic Stem Cells (HSCs)

HSCs are multipotent cells that generate all types of blood cells and can be found in peripheral blood, bone marrow (which provides an opportunity for autologous transplantation), and umbilical cord blood. Human HSCs (hHSCs) are one of the earliest cell types that have been used for clinical transplantation for the treatment of certain cancerous diseases, with acceptable safety [185]. Due to their ability to generate new blood and immune cells, hHSCs, especially autologous hHSCs, have also been used to treat autoimmune diseases of the nervous system, including MS, in the last two decades [186]. Currently, thousands of MS patients have been treated with autologous hHSCs, and many of them have shown significant therapeutic outcomes. However, in comparison to other stem cells, hHSCs only work as a supportive blood product following chemotherapy to speed hematopoietic recovery rather than a single treatment for these diseases. Actually, “autologous hematopoietic stem cell transplantation (AHSCT)” has become a clinical term that contains several procedures, including (1) mobilization—releasing hHSCs from the bone marrow into peripheral blood, (2) harvesting—collecting hHSCs from the blood of the patient, (3) conditioning regimen—administration of cytotoxic chemotherapy, and (4) infusion—returning hHSCs to the patient by infusion into the veins. Overall, with more clinical experience, AHSCT has become a promising supportive strategy for treating MS. The current status of using AHSCT for MS is well reviewed in a published paper [187].

## 4. Discussion

Cell therapy for neurological disorders has attracted more and more attention from researchers due to its unique and promising therapeutic potentials through the capabilities of cell replacement, neuroprotection, and promotion of intrinsic neuro-restoration. In this paper, we have introduced some promising cell types and their applications in treating neurological disorders both in animal studies and clinical trials. As we have summarized (see Table 7), each type of cell has its unique properties and may have different therapeutic functions in treating different diseases. However, it is worth mentioning that, although we have listed some common neurological disorders, the potential targets of the cells are not limited to what we have mentioned above. For example, MSCs have also been shown to be therapeutic for autism and meningitis, etc. [188,189]. An ideal cell type is not only about its therapeutic effects but also about its accessibility and cost and the time to obtain sufficient quantities. Therefore, it is hard to identify the best cell candidate for neurological disorders at the current stage without enough comparative data.

Aside from cell type, cell source, and the target disease, the outcome of cell therapy can also be highly influenced by other factors, including the route and cell quantity of cell administration. Intracerebral and intraspinal administration can bring more cells to the target position but may cause unexpected adverse effects due to the procedure, while intravenous or intranasal injections are safer but may lead to a loss of the cells. In addition, a successful cell therapy usually needs millions of cells for transplantation to humans; however, the cells are not “the more, the better” as more cells may result in negative outcomes, for example, reducing the safety of the cell therapy. Therefore, a proper delivery approach and a safe and therapeutic range of the quantity of the cells need to be found for specific diseases in future studies. Luckily, despite being at the pre-clinical stage of development, several methods have been proven to improve the therapeutic effects of the cells, such as genetic modification, pre-conditioning, and co-transplantation. With more studies conducted, it is possible to develop more effective cell therapies for patients with neurological disorders.

However, it is noteworthy that although animal studies usually show significant improvement after cell transplantation, the outcomes from clinical trials are normally not as suitable as from animal studies due to the differences between clinical diseases and animal models, such as the far greater heterogeneity between human patients than that of purpose-bred animals. Therefore, some success in animal studies may not be transferred to human studies, indicating the need to build up more clinically related animal models.

## 5. Conclusions

In this paper, we have introduced some promising cells that may be clinically therapeutic for the treatment of neurological disorders, based on their characteristics and the results from related clinical trials and recent animal studies. Overall, with more knowledge about these cells, it can be foreseen that cell therapy will have a crucial place in the future clinical management of neurological disorders, although there is still much work that remains to be done.

## Figures and Tables

**Table 1 biology-10-01142-t001:** Differences between autologous transplantation and allogeneic transplantation.

	Cell Source	Additional Invasive Procedures	Immunogenicity	Ethical Issues	Cell Availability	Cell Type
Autologous transplantation	Patients themselves	Yes	No immune rejection	No	Limited by autologous cell culture	Limited, depending on the patient him/herself
Allogeneic transplantation	Other donors	No	Activated immune response, immunosuppressants required	Yes (when it involves the use of human embryos)	Cryopreserved stocks, suitable for big amount cell preservation	Various, depending on the donors (e.g., ESCs and FSCs)

**Table 2 biology-10-01142-t002:** Examples of hESC transplantation for neurological disorders in clinical trials (data from PubMed).

Disease	Route of Administration	Cell Source	Cell Amount	Number of Patients	Longest Follow-Up Time (after 1st Transplantation)	Outcome/Conclusion	Ref.
Lyme disease	Intramuscular, intravenous, and other supplemental routes	Human embryo	N/A	59	8 weeks	43 patients showed significant improvement, 12 patients showed moderate improvement, 4 patients exhibited mild improvement in their brain perfusion; no deterioration was found	[29]
Lyme disease and multiple sclerosis	Intramuscular, intravenous, and other supplemental routes	Human embryo	N/A	2	N/A	Patients showed remarkable neurological functional and histological improvement; no adverse events were reported	[30]
Spinal cord injury	Intramuscular, intravenous, and other supplemental routes	Human embryo	hundreds of millions of cells in total	5	5 years	All patients showed neurological functional improvement, 3/5 showed improved American Spinal Injury Association score (ASIA); no adverse events were reported	[31]
Spinal cord injury	Intramuscular, intravenous, and other supplemental routes	Human embryo	hundreds of millions of cells in total	226	N/A	70% of patients improved by at least one ASIA grade after 3 phases of treatment; no adverse events were reported	[32]
Stroke	Intramuscular, intravenous, and other supplemental routes	Human embryo	hundreds of millions of cells in total	24	N/A	A large population of patients saw significant improvement regarding Nutech Functional Score and European Stroke Scale; no adverse events were reported	[33]
Multiplesclerosis	Intramuscular, intravenous, and other supplemental routes	Human embryo	hundreds of millions of cells in total	24	Around 1 year	Patients showed an improvement in parameters associated with MS when evaluated with reverse nutech functional score but not with the expanded disability status scale; no adverse events were reported	[27]
Cerebral palsy	Intramuscular, intravenous, and other supplemental routes	Human embryo	hundreds of millions of cells in total	91	N/A	Most patients showed significant improvement in Gross Motor Function Classification Scores Expanded and Revised (GMFCS-E & R)	[34]

**Table 3 biology-10-01142-t003:** Examples of hNSPC transplantation for neurological disorders in clinical trials (data from PubMed).

Disease (Model)	Route of Administration	Cell Source	Cell Amount	Number of Patients	Longest Follow-Up Time (after 1st Transplantation)	Outcome/Conclusion	Ref.
Ischemic stroke	Intracerebral	Human fetal spinal cord, cell line: NSI-566	1.2 × 10^7^, 2.4 × 10^7^, or 7.2 × 10^7^	9	24 months	All patients showed significant behavioral and histological improvements	[69]
ALS	Intraspinal	Human fetal brain	2.25–4.6 × 10^6^	18	51 months	No serious adverse effects. Some patients showed temporary subjective clinical improvement	[80]
ALS	Intraspinal	Human spinal cord	2 to 16 million	15	9 months	Intraspinal transplantation of human spinal cord-derived neural stem cells can be safely accomplished at high doses	[81]
ALS	Intraspinal	Human fetal spinal cord, cell line: NSI-566RSC	1.5 million	15	30 months	This NSPCs line can be safely transplanted into both lumbar and/or cervical human spinal cord segments	[67]
moderate PD	Intracerebral	Human fetal brain	2 million	7	4 years	No adverse effects; enhanced midbrain dopaminergic activity; minor neuropsychological changes; 6/7 showed improved motor function; 5/7 showed better response to medication	[82]
Chronic cervical SCI	Intraspinal	Cells were prepared and released by StemCells Inc.	15 to 40 million	16	1 year	Cell transplantation was safe, feasible, and well tolerated.Trends toward improvement in motor function and spasticity were seen	[83]
completethoracic SCI	Intraspinal	Human fetal spinal cord, cell line: NSI-566	N/A	4	27 months	No serious adverse events; 3/4 showed early signs of potential efficacy	[68]
Chronic cervical and thoracic SCI	Intraspinal	Human fetal brain, cell line: HuCNS-SC	20 to 40 million	29	1 year	Cell transplantation was safe and feasible using a manual injection technique	[84]
Traumatic cervical SCI	Intraspinal	Human fetal telencephalon	1 × 10^8^	34	1 year	No serious adverse effects, 5/19 of treated patients showed functional recovery, 1/15 untreated patients showed functional recovery	[85]
MS	Intrathecal	Autologous MSCs	3 × 10^7^	20	Around 1 year	No serious adverse effects, improved median Expanded Disability Status Scale (EDSS), 70% and 50% of the subjects demonstrated improved muscle strength and bladder function, respectively	[73]
MS	Intrathecal	Autologous bone marrow MSCs	0.08–17.6 million	6	8.9 years	No serious adverse events; 4/6 showed a measurable clinical improvement	[71]
non-acute severe TBI	Intravenous or intrathecal	Autologous MSCs	20 to 40 million	10	6 months	No serious adverse events, 7/10 patients presented different degrees of improvement in neurological function	[72]
Cerebral palsy	subarachnoid cavity	Autologous bone marrow MSCs	1–2 × 10^7^	60	6 months	No serious adverse events. Treated group showed significant motor function recovery but no significant increases in the language quotients	[70]

**Table 4 biology-10-01142-t004:** Examples of neuron transplantation for neurological disorders in clinical trials (data from PubMed).

Cell Type	Disease (Model)	Route of Administration	Cell Source	Cell Amount	Number of Patients	Longest Follow-Up Time (after 1st Transplantation)	Outcome/Conclusion	Ref.
sympathetic neurons	PD	Intracerebral	Autologous sympathetic neurons	N/A	4	36 months,	Clinical evaluations showed that an increase in the duration of levodopa-induced ‘‘on’’ phase, and the percent time spent in ‘‘off’’ phase exhibited a 30–40% reduction as compared to the pre-grafting values	[105]
Dopamine neuron-contained tissue	PD	Intracerebral	Human embryonic mesencephalic tissue	N/A	40	1 year	human embryonic dopamine neuron transplants survive in patients with severe Parkinson’s disease and result in some clinical benefit in younger but not in older patients	[106]
neuronal cells	Stroke	Intracerebral	Human teratocarcinoma	N/A	12	18 months	No adverse cell-related serologic or imaging-defined effects. The total European Stroke Scale score improved in six patients (3 to 10 points), with a mean improvement of 2.9 points in all patients	[107]

**Table 5 biology-10-01142-t005:** Examples of hMSCs transplantation for neurological disorders in clinical trials (data from PubMed).

Disease (Model)	Route of Administration	Cell Source	Cell Amount	Number of Patients	Longest Follow-Up Time (after 1st Transplantation)	Outcome/Conclusion	Ref.
Cerebellar ataxia	Intrathecal	Bone marrow, (cell line: CS20BR08)	2 × 10^6^/kg	1	10 months	No adverse events reported. Improved K-SARA (Korean version of the Scale for the Assessment and Rating of Ataxia) scores	[128]
ALS	Intrathecal	Autologous bone marrow	30 × 10^6^	8	14 months	No change in progression rate in patients with an inherently slow course, but some decreased progression rate in patients with an inherently rapid course	[129]
SCI	Intrathecal	Umbilical cord	4 × 10^6^/kg	143	12 months	No serious adverse events reported. Significant improvements in neurological dysfunction and recovery of quality of life	[130]
Acute complete SCI	Intraspinal	Umbilical cord	40 million	40	12 months	Promoted recovery of neurological function	[131]
Chronic SCI	Intradural and intravenous	Autologous bone marrow	6.6–7.6 × 10^7^	1	5 years	No complication or serious adverse effects, improved motoric function	[132]
Acute complete SCI	Intraspinal	Umbilical cord	4 × 10^7^	2	1 year	No obvious adverse symptoms reported, the supraspinal control of movements below the injury was regained by functional NeuroRegen scaffolds implantation with hMSCs	[133]
SCI	Intrathecal	Autologous adipose	9 × 10^7^	14	8 months	No serious adverse events. Several patients showed mild improvements in neurological function	[134]
Cerebral palsy	Intravenous	Umbilical cord	4.5–5.5 × 10^7^	39	13 months	hMSCs transplantation was safe and effective at improving the gross motor and comprehensive function of children with cerebral palsy when combined with rehabilitation	[135]
Cerebral palsy	Intrathecal and intravenous	Umbilical cord	8 × 10^6^/kg	1	18 months	No serious adverse effects reported. hMSCs transplantation improved functional recovery, combined with rehabilitation	[136]
Cerebral palsy	Intravenous	Umbilical cord	80 × 10^7^	1	5 years	hMSCs transplantation with basic rehabilitation improved the motor and comprehensive function. No serious adverse events	[137]
TBI	Intrathecal, intramuscular, and intravenous	Wharton’s jelly	18 × l0^6^ /kg	1	12 months	No important negative effects were reported. The patients’ speech, cognitive, memory, and fine motor skills were improved	[138]
Chronic ischemic stroke	Intracerebral	Bone marrow, cell line: SB623	2.5–10 × 10^6^	18	24 months	All experienced at least 1 treatment-emergent adverse event. 7 experienced 9 serious adverse events, which resolved without sequelae. Improved clinical outcomes	[139]
Chronic stroke	Intravenous	Autologous bone marrow	5.3 × 10^5^–2.9 × 10^6^/kg	9	60 weeks	No adverse event reported. Improved neurological functions and clinical outcomes	[140]
Stroke	Intravenous	Autologous bone marrow	0.6 to1.6 × 10^8^	12	12 months	No significant adverse effects were found. Improved neurological function	[141]
HIE	Intrathecal, intramuscular, and intravenous	Wharton’s jelly	12 × 10^6^/kg	1	12 months	Improved neurological recovery	[142]
HIE	Intravenous	Umbilical cord	1 × 10^8^	22	180 days	No significant adverse effects were found. Markedly improved recovery of neurological function, cognition ability, emotional reaction, and extrapyramidal function	[143]
MS	Intravenous	Umbilical cord	14 × 10^7^	20	1 year	No serious adverse events reported. Improved functional recovery	[144]
MS	Intravenous	Autologous bone marrow	1–2 × 10^6^/kg	24	6 months	No serious adverse effects reported. No substantial evidence of inhibition of disease activity, tissue repair, or recovery of function	[145]
MS	Intravenous	Umbilical cord	12 × 10^6^/kg	23	12 months	No significant adverse effects were found. Improved neurological function	[119]
Drug-resistant epilepsy	Intrathecal	Autologous bone marrow	7.4–16 × 10^7^	4	2 years	CD271+ hMSCs, combined with autologous bone marrow nucleated cells transplantation, showed no serious adverse events but considerable neurological and cognitive improvement	[146]
PD	Intracerebral	Autologous bone marrow	N/A	7	36 months	No significant adverse effects were found. Several patients showed improved neurological function	[147]

**Table 6 biology-10-01142-t006:** Examples of OECs transplantation for neurological disorders in clinical trials (data from PubMed).

Disease	Route of Administration	Cell Source	Cell Amount	Number of Patients	Longest Follow-Up Time (after 1st Transplantation)	Outcome/Conclusion (Targeting Behavioral and Histological Change)	Ref.
SCI	Intraspinal	Autologous olfactory bulb	5 × 10^5^	1	19 months	Improved neurological and histopathological recovery, no adverse effects were seen	[178]
SCI	Intraspinal	Human fetal olfactory bulbs	1 × 10^6^	7	6 months	No serious adverse effects were seen. All treated patients showed functional improvement, 4/5 showed improvement in electrophysiological tests	[177]
SCI	Intraspinal	Human fetal olfactory bulbs	2–5 × 10^6^	15	8 weeks	No serious adverse effects were seen. 12/15 showed obvious spinal function improvement, and 3/15 had slight improvement	[179]
SCI	Intraspinal	Autologous nasal mucosa	1.8–21.2 × 10^6^	6	1 year	no adverse findings related to olfactory mucosa biopsy or transplantation. All treated patients showed improved functional recovery, 2/3 of treated patients showed improved American Spinal Injury Association class	[176]
SCI	Intraspinal	Human fetal olfactory bulbs	2 × 10^6^	6	24 months	No clinical complications were observed. All patients showed improved neurofunctional recovery	[180]
SCI	Intraspinal	Autologous olfactory mucosa	Not mentioned	8	24 months	No clinical complications were observed. All patients showed improved neurofunctional recovery, 3/8 showed returned substantial sensation and motor activity in various muscles, 2/8 showed restored bladder function	[181]
SCI	Intraspinal	Human fetal olfactory bulbs	5 × 10^5^	11	1.5 years	All patients had no complications or deterioration of neurological conditions. Sensation and spasticity improved moderately. Locomotion recovery was minimal	[182]
SCI	Intraspinal	olfactory bulbs	5 × 10^5^	108	5.3 years	No serious adverse effects were seen. Improve neurological functions. Sufficient rehabilitation most likely played a critical role	[183]
SCI	Intraspinal	olfactory bulbs	5 × 10^5^	171	12 weeks	OECs transplantation can improve the neurological function of spinal cord of SCI patients regardless of their ages	[184]
ALS	Intracranial and/or intraspinal	Human fetal olfactory bulbs	1–2 × 10^6^/treatment, 1–5 treatments	507	N/A	multiple doses of cellular therapy serve a positive role in the treatment of ALS	[173]
Cerebral palsy	Intracranial	Human fetal olfactory bulbs	2 × 10^6^	14	6 months	OECs transplantation is effective for functional improvement in children and adolescents with cerebral palsy, yet without obvious side effects	[174]

**Table 7 biology-10-01142-t007:** Characteristics of cell candidates for neurological disorders through direct transplantation.

Cell Type	Stemness	Advantage	Disadvantage	Examples of Targeted Neurological Disorders in Animal Studies	Examples of Targeted Neurological Disorders in Clinical Trials
hESCs	Pluripotent	Unlimited proliferation	Ethical issues; risk of immune rejection, risk of tumor formation	SCI	Lyme disease, MS, SCI, stroke, cerebral palsy
hiPSCs	Pluripotent	No ethical issues; applicable for autologous transplantation; high accessibility	Risk of tumor formation, unpredictable mutagenesis	SCI, stroke	N/A
hNSPCs	Multipotent	Neural lineage differentiation; low risk of tumor formation; multiple sources	Comparatively low proliferation	SCI, HD, stroke, PD, AD, ataxia, TBI, ALS	Stroke, ALS, PD, SCI, MS, TBI, cerebral palsy
Neurons	Terminal cells	No risk of tumor formation; no unexpected differentiation	Poor survival after transplantation	SCI, ALS, PD, AD	PD, stroke
Oligodendrocytes	Terminal cells	No risk of tumor formation; no unexpected differentiation	Poor survival after transplantation	MS, SCI	N/A
Astrocytes	Terminal cells	No risk of tumor formation; no unexpected differentiation	Effects highly depend on the subtype of the cells; not much studied	SCI	N/A
hMSCs	Multipotent	Applicable for autologous transplantation; high accessibility; low risk of tumor formation	Limited neural differentiation; effects may be not as suitable as hNSPCs	SCI, PD, stroke, TBI, ALS, ataxia, MS, AD, epilepsy	SCI, PD, stroke, TBI, ALS, ataxia, MS, epilepsy, cerebral palsy, HIE
DPSCs	Multipotent	Applicable for autologous transplantation; high accessibility; low risk of tumor formation	Limited neural differentiation; high heterogeneity; low number of cells from pulp tissue	SCI, HD, ataxia, stroke, PD	N/A
SHED	Multipotent	High accessibility; low risk of tumor formation	Limited neural differentiation; high heterogeneity; low number of cells from pulp tissue	SCI, stroke	N/A
Muse cells	Pluripotent	Applicable for autologous transplantation; high accessibility; non-tumorigenicity	Not much studied	SCI, stroke, HIE, ALS	N/A
hOECs	Terminal cells	Applicable for autologous transplantation; high accessibility; non-tumorigenicity; no unexpected differentiation	Hard to purify; poor survival after transplantation; limited migration and phagocytosis;	SCI	SCI, ALS, cerebral palsy
hHSCs	Multipotent	Applicable for autologous transplantation; high accessibility	Some risk of serious adverse effects	N/A	MS

Abbreviation: SCI = spinal cord injury; MS = multiple sclerosis; HD = Huntington’s disease; TBI = traumatic brain injury; PD = Parkinson’s disease; AD = Alzheimer’s disease; ALS = amyotrophic lateral sclerosis; HIE = hypoxic-ischemic encephalopathy; N/A = not applicable.

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
