# Peer review of "Cell Therapy for Neurological Disorders: The Perspective of Promising Cells"

_biology, 2021, doi:10.3390/biology10111142_

Round 1

Reviewer 1 Report

In the article “Cell therapy for neurological disorders: the perspective of promising cells”, authors provide a comprehensive review of the various promising cell types that may be clinically used for many neurological disorders, based on their characteristics and the progress in related studies. Though the review written by authors is organized and very informative, I would like to comment on the overall structure and on some minor corrections that need to be addressed in the manuscript.

  1. Authors can expand a little more on the section 2- Cell source: autologous or allogeneic. This may include the advantages and challenges of these two cell sources. Authors can compare these two possible cell sources and mention briefly about their advantages and disadvantages with respect to its immunogenicity, cell stability, Preservation/production/modification of cells, toxicity, ethical issues etc.
  2. In section 3, It would be great if authors can do some rearrangement while they list different types of cells and their current progress. Authors can classify the whole section into major three sub categories which comprise of-Stem cells, iPSC and differentiated cells. And, they can include different cell types under this major subclass. While you change into a new order, please make sure each section is advancing well with a proper connection in terms of its promises/advantages and challenges in the field. This rearrangement would be helpful from a reader’s perspective.
  3. Change the font style of in vitro/in vivo to italics in the manuscript.
  4. There are many typos in the manuscript (Eg-line 182-To date?). Please correct them wherever it is necessary in the whole manuscript.

Reviewer 2 Report

As the autors claim, this review aims to introduce the potentials of 
different cells for treating neurological disorders by collecting the results from clinical trials and animal studies since 2016. They also claim that the characteristics of each cell type and the main "mechanism of function" are described. Unfortunately, I do not see an updated and relevant characterization of the neural stem and progenitor cells or the differentiated cells making the CNS and affected in different neurological diseases, as well as the stategies used to replace them by transplanted cells.  Instead, many non neural cell are presended as options, but far away to point to their mechanisms in the respective diseases. While trying to cover a large field, this rewiew does not provide a critical judgement and does not discuss the directions to be followed to advance the cell therapy in CNS.

Reviewer 3 Report

This study describes the characteristics of effective cell source on cell therapy for neurological disorders. The authors also properly cite the papers and clearly summarizes the important points.

I would have listed below some suggestions.

  1. Page1 Figure: Lack of figure legends. Please provide detailed descriptions.
  2. Page20 line 420: Although it is described as “3.6. Discussion”, it need to be changed considering the structure so far.
  3. Page 22 Table 6: This table is very important for cell transplantation progress on neurological areas. If possible, add original meaning for the abbreviation of neurological disorders.
  4. Page24 line 460: “examples of hNSPC” should be unified with other descriptions.(e.g. Table S1: Examples of hiPSC or Table S3:Examples of human neuron)

Reviewer 4 Report

In the present review, Donghui Liu et al. explore the potentials of different cell types in treating neurological disorders, by collecting the results from related clinical trials and animal studies. The manuscript is well written and exhaustive, and the tables make it easier to read.

Suggested minor changes:

Line 29-30 Graphical abstract is small. Please provide bigger image 

Lines 65-67 These last lines repeat what previously asserted. I suggest eliminating them.

Adipose-derived mesenchymal stem cells (A-MSCs) should be hADSCs, please modify in the text.

All acronyms must be explained the first time they appear in the text, see for example line 169 “6-OHDA”. What does it mean? Please explain.
